# Chemical Composition, Repellent Action, and Toxicity of Essential Oils from *Lippia origanoide*, *Lippia. alba* Chemotypes, and *Pogostemon cablin* on Adults of *Ulomoides dermestoides* (Coleoptera: Tenebrionidae)

**DOI:** 10.3390/insects14010041

**Published:** 2022-12-31

**Authors:** Karina Caballero-Gallardo, Katerin Fuentes-Lopez, Elena E. Stashenko, Jesus Olivero-Verbel

**Affiliations:** 1Environmental and Computational Chemistry Group, School of Pharmaceutical Sciences, Zaragocilla Campus, University of Cartagena, Cartagena 130014, Colombia; 2Functional Toxicology Group, School of Pharmaceutical Sciences, Zaragocilla Campus, University of Cartagena, Cartagena 130014, Colombia; 3National Research Center for Agroindustrialization of Aromatic Medicinal and Tropical Species (CENIVAM), Universidad Industrial de Santander, Bucaramanga 680002, Colombia

**Keywords:** insects, biopesticides, biodiversity, repellent, chemotype

## Abstract

**Simple Summary:**

Essential oils (EOs) from plants are promising products for pest management. This paper describes the chemical composition and repellent action of four EOs against *Ulomoides dermestoides*, a common pest on several stored products. Most abundant chemical components found in the EOs were sabinene, trans-β-caryophyllene, and α-humulene for phellandrene-rich *Lippia origanoides*; limonene and carvone for carvone-rich *Lippia alba*; geranial, geraniol, and neral for citral-rich *Lippia alba*; and α-guaiene, α-bulnesene, and patchoulol for *Pogostemon cablin*. The repellent bioactivity, carried out utilizing the area preference method, showed that all EOs displayed great repellency with low mortality rates, suggesting these natural mixtures can be used in formulations of repellents against stored grain pests.

**Abstract:**

The essential oils (EOs) from bioactive species can provide an alternative tool for the management of stored grain insects that is less environmentally damaging than synthetic chemicals. The aim of this study was to assess the repellent action and toxicity of EOs obtained from phellandrene-rich *Lippia origanoides*, carvone-rich *Lippia alba*, citral-rich *L. alba*, and *Pogostemon cablin* aerial parts on adults of *Ulomoides dermestoides*. These EOs were isolated by hydrodistillation and characterized by gas chromatography coupled to mass spectrometry (GC-MS). The repellency assay was carried out using the area preference method, and the toxicity evaluated utilizing a filter paper contact test. The major components (>10%) of the studied EOs were sabinene (16.9%), *trans*-β-caryophyllene (18.6%) and α-humulene (10.1%) for phellandrene-rich *L. origanoides* EO; limonene (40.1%) and carvone (37.7%) for carvone-rich *L. alba* EO; geranial (24.5%), geraniol (19.0%), and neral (11.9%) for citral-rich *L. alba* EO; and α-guaiene (13.3%), α-bulnesene (15.7%), and patchoulol (35.3%) for *P. cablin* EO. All EOs displayed 100% repellency at a concentration of 16 μL/mL, with lower toxicity than that elicited by the commercial repellent DEET. EO concentrations up to 8 µL/mL did not induce any mortality on the beetle. These findings show that the EOs provide active and safe molecules for natural repellent formulations to prevent and control insect infestations of stored products.

## 1. Introduction

About 30% of stored food products are highly vulnerable to post-harvest loss, both in terms of quality and quantity [1]. Insect pest infestations represent a major threat to these products, which in turn generate suitable conditions for the growth of bacteria and fungi [2].

The uncontrolled and excessive use of synthetic pesticides has caused serious environmental and health problems [3,4,5]. In order to avoid multiple post-harvest losses and guarantee food security, various strategies for pest control are being developed. The search for natural products is a good alternative to replace synthetic pesticides, since they are a rich source of secondary metabolites with biological activity, and beneficial properties with low toxicity in exposed organisms. Essential oils (EOs) are mixtures of small molecules that can be used as active ingredients, and are widely studied for the discovery of new bioactive compounds against pest insects. In fact, essential oil compounds such as 1,8-cineole and limonene have been used as active ingredients in mosquito repellents, flea shampoos, and insecticide-related formulations [6].

The peanut beetle, *Ulomoides dermestoides* Fairmaire (Coleoptera: Tenebridae), is considered a worldwide pest of stored grains (corn, oats, peanuts, rice, among others), which has been controlled with chemical insecticides. However, the use of these products has generated environmental contamination and insect resistance to these synthetic substances. Essential oils have proven to be friendly to the environment, due to their biodegradability, insect selectivity, and low toxicity to vertebrates [2,7,8], playing an important role in insect-plant interactions, and being part of defense strategies against herbivorous insects. Some of the components of EOs exert toxic effects on insects, either by contact, ingestion, or fumigation, as well as deterrence, inhibition of feeding, and repellency [9].

On the other hand, there is an interest in studying and evaluating the use of EOs for pest management, mainly as a result of insect resistance to the synthetic insecticides. In fact, various studies have investigated the bioactivity of EOs and their potential uses as bio-insecticides against insect pests. In this sense, EOs from *Lippia origanoides* (phellandrene)*, L. alba* (carvone), and *Pogostemon cablin* have shown repellent and insecticidal properties against *Aedes aegypti* [10], *Sitophilus zeamais* [11], and *Tribolium castaneum* [11,12].

The present study aimed to determine the chemical composition of the essential oils from phellandrene-rich *L. origanoides*, carvone-rich *L. alba*, citral-rich *L. alba*, and *P. cablin,* and evaluate their repellent action and toxicity on adults of *U. dermestoides*.

## 2. Materials and Methods

### 2.1. Insects

Adult insects of the beetle *Ulomoides dermestoides* (Coleoptera: Tenebrionidae) were housed in glass jars covered with a nylon mesh held with rubber bands, under standard laboratory conditions of 28 °C (±1 °C), a 10:14 h light:dark photoperiod, and relative humidity of 70–85%. Insect maintenance, feeding, and breeding followed the established laboratory protocols in the bioassay laboratory of the University of Cartagena (Colombia) [13]. Adult, healthy *U. dermestoides*, 4-8 days old, were randomly chosen for bioassays.

### 2.2. Plant Material and Extraction of EOs

EOs of *Lippia origanoides*, phellandrene-rich chemotype, *Lippia alba*, carvone-rich chemotype, *Lippia alba*, citral-rich chemotype, and *Pogostemon cablin* were obtained from aerial parts using steam distillation in a 0.4 m^3^ stainless steel column and separated by decantation. Subsequently, they were dried with Na_2_SO_4_ and stored at 4 °C in amber flasks.

### 2.3. Characterization of EOs

Each EO (50 mg) was dissolved in 1 mL of CH_2_Cl_2_. The dilution (2 µL) was injected into a gas chromatograph coupled to a mass selective detector. The analysis was performed in a gas chromatograph, GC 6890 Plus (Agilent Technologies, Palo Alto, CA, USA), equipped with a selective mass detector, MS 5973 *Network* (AT, Palo Alto, CA, USA), using electron ionization (EI, 70 eV). Helium (99.995%, AP gas, Messer, Bogotá, Colombia) was used as carrier gas, with initial inlet pressure at the head of the column of 113.5 kPa; the volumetric flow rate of carrier gas during the chromatographic run was kept constant (1 mL/min). The injection mode was split (30:1) and the injector temperature was kept at 250 °C.

Chromatographic separation of EO components was carried out in two capillary columns, one with poly(ethylene glycol), PEG, as a stationary phase (DB-WAX, J & W Scientific, Folsom, CA, USA), 60 m × 0.25 mm (i.d) × 0.25 μm (df), and the other containing 5%-phenyl-poly(methylsiloxane) (DB-5MS, J&W Scientific, Folsom, CA, USA), displaying the same dimensions as the polar one. When using the polar column (DB-WAX), the oven temperature was programmed from 50 °C (5 min) to 150 °C (7 min), at 4 °C/min, and then up to 230 °C (50 min), at 4 °C/min. When the apolar column (DB-5MS) was used, temperature was programmed from 45 °C (5 min) to 150 °C (2 min), at 4 °C/min, then up to 300 °C (10 min), at 5 °C/min. The GC/MS transfer line temperature was set at 230 °C when the polar column was used and at 300 °C for the apolar column. The temperatures of the ionization chamber and the quadrupole were 250 °C and 150 °C, respectively. The mass range for the acquisition of ionic currents was *m/z* 45–450 u, with an acquisition speed of 3.58 scan/s. The data was processed with the MSD Chem Station software (AT, Palo Alto, CA, USA). The integration parameters were the following: threshold = 18, with a rejection area of the peak above the baseline less than 1%. The identification of compounds was carried out based on their linear retention indices (LRI), calculated from the retention times of the compound of interest, and the C6–C25 and C8–C40 n-alkanes (Sigma-Aldrich, San Luis, MO, USA), according to the used temperature (Equation (1)):LRI = (100 × n) + 100 × [(t_Rx_ − t_Rn_)/(t_RN_ − t_Rn_)](1)
where LRI is the linear retention index of the compound of interest (x), n is the carbon number of the n-alkane that elute first, and N the carbon number of the n-alkane that elutes after the x; t_Rx_ is the retention time of the compound of interest (min), and t_Rn_ and t_RN_ are the retention times of n-alkanes that elute before (n) or after (N) the compound of interest (x) (min), respectively.

For tentative identification, experimental mass spectra of each compound was compared to that from Adams [14], NIST [15], and Wiley spectral databases. Confirmatory identification of some detected compounds was performed by comparing their LRI and mass spectra with those of available standard substances.

### 2.4. Repellent Action of EOs

The experiments were carried out in Petri dishes (9 cm in diameter) employing the area preference method on filter paper [16] under standard laboratory conditions of 28 °C (±1 °C), a 10:14 h light:dark photoperiod, and relative humidity of 70–85%. Briefly, each paper filter was cut in two identical pieces, one (left) treated with 500 µL acetone (negative control), and the other (right) with 500 µL of 2–16 μL/mL of each EO dissolved in acetone. The solvent was allowed to evaporate for 10 min, and then the two halves were re-attached with adhesive tape. Ten unsexed adults of *U. dermestoides* were deposited in the center of the filter paper and the Petri dish was closed with its fitting cover and stored in the absence of light. After exposure (2 and 4 h), experimental units were counted in both areas. A commercial repellent containing 99.8% DEET was utilized as a positive control. The Percentage of Repellency (PR) and Preference Index (PI) were defined as PI = (percentage of insects in treated paper − percentage of insects in control paper)/(percentage of insects in treated paper + percentage of insects in control paper). PI values ranging between −1.0 and −0.1 indicate the EO has repellent properties; values from −0.1 to +0.1 suggest the EO has neither repellent nor attractant behaviors; and those between +0.1 and +1.0 make the EO an insect attractant. Each experiment was carried out twice, with four replicates each.

### 2.5. Contact Toxicity on Filter Paper

The contact toxicity on filter papers was conducted using Whatman grade 1 filter (Catalog number 1001090, Whatman International Ltd, Maidstone, UK), 9 cm in diameter [17]. One mL of acetone (Merck, Darmstadt, Germany) or treatment (EO in acetone, 2–16 µL/mL) was dispensed on the surface of the paper that was then placed in a glass Petri dish. DEET (99.8%) (WPC Brands, Inc, Bridgeton, MO, USA) was utilized as positive control. Once the solvent evaporated (10 min), ten unsexed adults were added on each disc, closed with a fitting cover and then stored in darkness under laboratory conditions. Mortality was assessed after 24 and 48 h. The insects were considered dead when no leg or antennal movements were observed. The experiment was repeated twice, using for replicates.

### 2.6. Data Analysis

The data are presented as the mean ± standard error (x ± SE). The paired *t*-test was utilized to compare mean number of insects on the treated and untreated area of the filter paper. Percentage repellency was preceded with a positive or negative sign, indicating repellency or attraction, respectively. Data obtained from each bioassay were subjected to Probit analysis, and 50% repellence concentrations (RC_50_) were determined by log-Probit regression. Normal distribution and homogeneity of variances were assessed using Shapiro–Wilk and Bartlett tests, respectively. Data from the assays were arcsine-transformed and subjected to ANOVA to determine differences between means of different treatments, employing Dunnett as a post-hoc test. Chi-square was applied to establish relationships between repellency and treatments. Calculations were carried out using GraphPad Prism 5.0 (GraphPad Software, Inc., San Diego, CA, USA). Significance was set at *p* < 0.05.

## 3. Results

### 3.1. Chemical Composition of EOs

The major compounds (relative amount ≥5%) found in phellandrene-rich *L. origanoides*, carvone-rich *L. alba*, citral-rich *L. alba*, and *P. cablin* EOs are listed in Table 1. In the phellandrene-rich *L. origanoides* EO, sabinene (16.9%), *trans*-β-caryophyllene (18.6%), α-humulene (10.1%), *p*-cimene (8.7%), and 1,8-cineol (6.5%) were the major constituents, representing 60.8% of the EO. The major components in the carvone-rich *L. alba* EO were limonene (40.1%), carvone (37.7%), and germacrene D (8.1%), accounting for 85.9% of the EO. In the *L. alba*, citral-rich chemotype EO, geranial (24.5%), geraniol (19.0%), neral (11.9%), and *trans*-*β*-caryophyllene (9.1%), were the most abundant, accounting for 64.5% of the EO. The major compounds in the *P. cablin* EO were patchoulol (35.3%), α-bulnesene (15.7%), α-guaiene (13.3%), seychellene (8.5%), and α-patchoulene (6.3%), adding up 79.1% of the EO. Chromatographic profiles for testes EOs, as well as the mass spectra for most identified compounds, are provided in Appendix A.

### 3.2. Repellent Action on U. dermestoides Adults

The repellent effects of the EOs and the commercial repellent DEET on adult *U. dermestoides* are displayed in Table 2. The EOs from *L. alba*, carvone and citral chemotypes, as well as the EO extracted from *P. cablin* were strongly repellent to *U. dermestoides*. These EOs exhibited PR greater than 80% at the lowest concentration (2 μL/mL) after 2 h exposure. In the four evaluated EOs, no attractive action was found, in contrast with that observed for DEET at the lowest concentration after exposure to 2 h (−13 ± 11) and 4 h (−25 ± 13). The commercial repellent displayed less efficacy than the four tested EOs, with PR lower than 100% at the highest tested concentration (16 μL/mL, 2 h exposure). Interestingly, all four EOs were found to be more repellent (RC_50_ 1.0–4.3 μL/mL) than DEET (RC_50_ 5.9–9.1 μL/mL) under the same experimental conditions. Significant differences at all tested concentrations in *L. alba*, carvone and citral chemotypes, and *P. cablin* EOs, as well as commercial repellent DEET were registered between the average number of insects in the treated and untreated areas. In addition, no statistical differences were detected between PRs when the same tested concentrations of EOs were compared at 2 and 4 h exposure.

The preference index calculated for tested EOs are shown in Table 3. All EOs displayed PI values between −1.0 and −0.1, indicating that these EOs are repellents against *U. dermestoides* adults. This same pattern was found in the commercial repellent DEET except for the lowest concentration (2 μL/mL) where the PI had an attractant effect.

### 3.3. Contact Toxicity

The results of contact toxicity are presented in Figure 1. Compared to DEET, the four evaluated EOs exhibited lower toxicity against *U**. dermestoides*. The toxicity decreased in the order carvone-rich *L. alba* > citral-rich *L. alba*> phellandrene-rich *L. origanoides* > *P. cablin*. In contrast, the commercial repellent caused more than 80% mortality at the lowest tested concentration (2 μL/mL) at both 24 and 48 h exposure. An interesting point displayed in Figure 1 is that the biological activity of the EOs was more variable than that elicited by DEET.

## 4. Discussion

Essential oils and their components possess different bioactivities that have been employed for pharmacological, medicinal, aromatic or cosmetic purposes. In addition, they are considered to be an alternative for the control of insect pests of stored products [21]. In this study, several EOs were chemically characterized by GC-MS and tested their repellent and toxicity properties against *U. dermestoides* under laboratory conditions.

EOs from *L. alba* (carvone and citral chemotypes) exhibited a high repellency action against *U. dermestoides* adults. These results were similar to those reported for others insects, such as *Sitophilus zeamais* [11], *Tribolium castaneum* [11], and *Rhipicephalus microplus* [22]. Species of the genus *Lippia* (Verbenaceae) are characterized by their wide distribution and medicinal importance. In Colombia, *L. origanoides* species have been reported to have properties such as antioxidant [23], repellent [24], antifungal [25], and fumigant [13]. Moreover, *L. alba* species have shown antifungal, antigenotoxic [26], antibacterial [27], and repellent properties [28], among others.

The repellent action of the different *L. alba* chemotypes is a function of their composition. In this study, most abundant compounds present in the *L. alba* EOs, such as limonene, carvone, geranial, geraniol, and neral have been evaluated as repellents against different insect pest of stored products [28,29,30]. In the case of carvone and geraniol, main compounds present in *L. alba*, displayed repellency percentages greater than 90% against *Tribolium castaneum* [28]. In addition, Zhang et al. [31] reported that geraniol was a strong repellent on the booklouse, *Liposcelis bostrychophila,* and limonene showed great repellent activity against *T. castaneum* and *Lasioderma serricorne*.

*Pogostemon cablin*, also known as “patchouli”, is an aromatic plant member of the Lamiaceae family, widely cultivated in many tropical and subtropical regions. The main compounds found in *P. cablin* oil were similar to those reported by other authors, with relatively small differences in concentrations, reaching a maximum of 2.1% in the case of α-guaiene [32,33]; 3.8% for α-bulnesene [32]; 2,7% for α-patchoulene [32]; 4.8% for seychellene [33]; and 12.7–15.8% for patchoulol [32,34]. This EO has also been tested as a repellent in *Tribolium castaneum*, *Lasioderma serricorne*, and *Liposcelis bostrychophila* [34] with moderate results, suggesting a broader-spectrum use, in particular combined with other potent EOs, such as those from *Lippia* species.

According to Peixoto et al. [11] carvone chemotypes were more toxic than the citral chemotypes against *Sitophilus zeamais* and *Tribolium castaneum,* a behavior similar to that presented here. Interestingly, compared to DEET, EOs evaluated in this research showed lower toxicity against adult *U. dermestoides*. This has a two-fold consequence. First, the EOs from species reported here are more environmentally friendly from a contact toxicity perspective; and second, their combined use guarantees the presence of multiple molecules that can alleviate possible resistance-related problems linked to the use of individual compounds when used as repellents.

## 5. Conclusions

The EOs from *L. alba* carvone and citral chemotypes, as well as that from *P. cablin* have great potential to act as repellents on *U. dermestoides*, with lower toxicity compared to DEET. The combination of these EOs in formulations of environmentally friendly repellents is highly encouraged.

## Figures and Tables

**Figure 1 insects-14-00041-f001:**
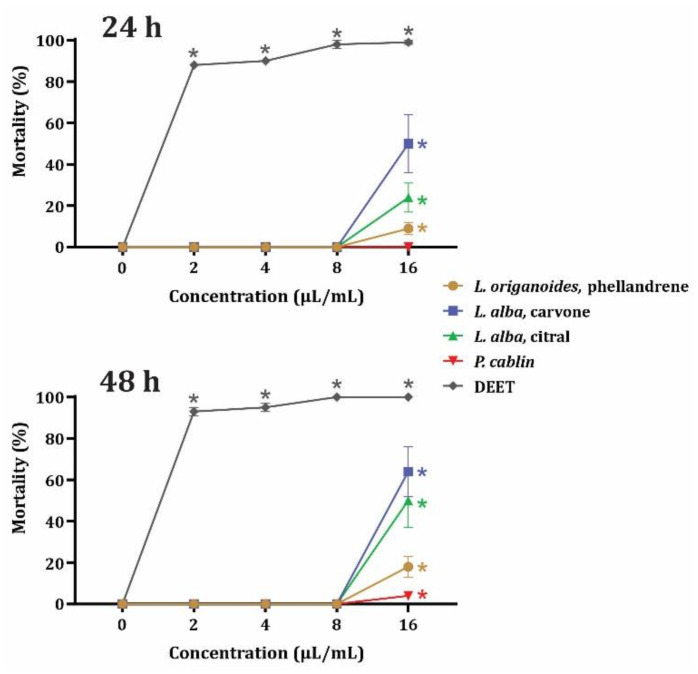
Percentage mortality of *U. dermestoides* adults exposed to EOs from phellandrene-rich *L. origanoide*, carvone-rich *L. alba*, citral-rich *L. alba*, and *P. cablin,* after 24 and 48 h exposures. Data (n = 8) are presented as mean ± SEM. *. Significant difference compared to the negative control.

**Table 1 insects-14-00041-t001:** Chemical composition of phellandrene-rich *L. origanoides*, carvone-rich *L. alba*, citral-rich *L. alba*, and *P. cablin* essential oils.

Compound	Linear Retention Indices (LRI)				GC Relative Area, % DB-5MS				Identification Method
	DB-5MS		DB-WAX						
	LRI_exp_	LRI_ref_	LRI_exp_	LRI_ref_	*L. origanoides*, Phellandrene Chemotype ^a^	*L. alba*, Carvone Chemotype ^b^	*L. alba*, Citral Chemotype ^c^	*P. cablin* ^d^	
*cis*-Hex-3-en-1-ol	853 ^b,c^	850 ^b,c,1^	1386 ^b,c^	1380 ^b,c,2^	ND	0.1	0.2	ND	e, f
α-Pinene	938 ^a^, 935 ^b,c,d^	932 ^a,b,c,d,1^	1022 ^a,b,d^ 1023^c^	1025 ^a,b,c,d,2^	2.0	0.1	0.3	0.1	e, f, g
Camphene	955 ^a^, 951^b^	954 ^a,b,1^	1066 ^a,b^	1069 ^a,b,2^	0.5	0.2	ND	ND	e, f, g
Sabinene	976 ^a^	975 ^a,1^	1120 ^a^	1122 ^a,2^	16.9	ND	ND	ND	e, f, g
Oct-1-en-3-ol	978 ^b,c^	980 ^b,c,2^	1452 ^b,c^	1444 ^b,c,2^	ND	0.1	0.2	ND	e, f, g
β-Pinene	982 ^a^, 980 ^d^	974 ^a,1;d,3^	1108 ^a^, 1109 ^d^	1110 ^a,d,2^	2.5	ND	ND	0.1	e, f
β-Myrcene	990 ^a,b^	988 ^a,1^, 990 ^b,1^	1162 ^a^, 1163 ^b^	1161 ^a,b,2^	0.4	0.9	ND	ND	e, f, g
Octan-3-ol	994 ^b^	991 ^b,1^	1396 ^b^	1392 ^b,2^	ND	0.1	ND	ND	e, f
α-Phellandrene	1011 ^a^, 1005 ^c^	1002 ^a,c,1^	1163 ^a^, 1162 ^c^	1168 ^a,c,2^	3.6	ND	0.1	ND	e, f, g
*p*-Cymene	1029 ^a^	1024 ^a,1^	1267 ^a^	1270 ^a,2^	8.7	ND	ND	ND	e, f, g
Limonene	1033 ^a^, 1032 ^b,c^	1029 ^a,b,c,1^	1197 ^a^, 1199 ^b^, 1201 ^c^	1198 ^a,b,c,2^	4.4	40.1	2.4	ND	e, f, g
β-Phellandrene	1037 ^a^	1025 ^a,1^	1206 ^a^	1209 ^a,2^	3.1	ND	ND	ND	e, f
1,8-Cineole	1036 ^a^	1031 ^a,1^	1211 ^a^	1211 ^a,2^	6.5	ND	ND	ND	e, f, g
*trans*-β-Ocimene	1047 ^b,c^	1050 ^b,c,1^	1253 ^b,c^	1250 ^b,c,2^	ND	0.5	0.3	ND	e, f
γ-Terpinene	1062 ^a^	1059 ^a,1^	1244 ^a^	1245 ^a,2^	1.0	ND	ND	ND	e, f, g
Linalool	1100 ^a^, 1099 ^b,c^	1096 ^a,b,c,1^	1549 ^a,b,c^	1543 ^a,b,c,2^	0.5	0.7	1.1	ND	e, f, g
*trans*-*p*-mentha-2,8-dien-1-ol	1123 ^b^	1122 ^b,1^	1633 ^b^	1639 ^b,2^	ND	0.2	ND	ND	e, f
*cis-Limonene oxide*	1141 ^b^	1136 ^b,1^	1453 ^b^	1451 ^b,2^	ND	0.1	ND	ND	e, f, g
Borneol	1175 ^a^, 1174 ^b^	1169 ^a,b,1^	1700 ^a^, 1701 ^b^	1700 ^a,b,2^	1.8	0.7	ND	ND	e, f
Terpinen-4-ol	1186 ^a^	1177 ^a,1^	-	1601 ^a,2^	0.4	ND	ND	ND	e, f, g
*cis*-Dihydrocarvone	1200 ^b^	1192 ^b,1^	-	-	ND	0.2	ND	ND	e, f
*trans*-Dihydrocarvone	1208 ^b^	1200 ^b,1^	-	-	ND	0.5	ND	ND	e, f
*cis*-Carveol	1231 ^b^	1229 ^b,1^	1839 ^b^	1854 ^b,2^	ND	0.1	ND	ND	e, f
*neo*-*iso*-Dihydrocarveol	1242 ^b^	1228 ^b,1^	-	-	ND	0.1	ND	ND	e, f
Carvone	1259 ^b^	1258 ^b,3^	1746 ^b^	1734 ^b,2^	ND	37.7	ND	ND	e, f, g
Piperitone	1265 ^b^	1264 ^b,3^	1738 ^b^	1730 ^b,2^	ND	1.9	ND	ND	e, f
Geranial	1274 ^b,c^	1270 ^b,c,2^	1740 ^b,c^	1725 ^b,c,2^	ND	0.5	24.5	ND	e, f, g
*trans*-Carvyl acetate	1346 ^b^	1342 ^b,1^	-	1727 ^b,2^	ND	0.1	ND	ND	e, f
Piperitenone	1347 ^b^	1343 ^b,1^	1929 ^b^	1909 ^b,2^	ND	0.8	ND	ND	e, f
α-Cubebene	1356 ^a^	1351 ^a,2^	1462 ^a^	1460 ^a,2^	0.5	ND	ND	ND	e, f
α-Copaene	1386 ^a^, 1387 ^b^	1376 ^a,b,1^	1497 ^a,b^	1491 ^a,b,2^	1.1	0.1	ND	ND	e, f
β-Bourbonene	1395 ^b^	1384 ^b,2^	1524 ^b^	1523 ^b,2^	ND	3.0	ND	ND	e, f
β-Elemene	1397 ^a^, 1396 ^b,c^	1390 ^a,b,c,1^	1595 ^a^, 1594 ^b^, 1588 ^c^,	1591 ^a,b,c,2^	1.0	0.6	2.6	ND	e, f
trans-β-Caryophyllene	1435 ^a^, 1434 ^b,c,d^	1427 ^a,b,c,d,3^	1606 ^a^, 1602 ^b^, 1611 ^c^, 1612 ^d^	1599 ^a,b,c,d,2^	18.6	0.3	9.1	2.6	e, f, g
β-Copaene	1442 ^a^, 1433 ^b^	1433 ^a,2^, 1432 ^b,1^	1597 ^a,b^	1580 ^a,b,2^	1.0	0.3	ND	ND	e, f
β-Gurjunene	1443 ^b^	1433 ^b,1^	-	1597 ^b,2^	ND	0.3	ND	ND	e, f
*trans*-β-Farnesene	1456 ^b^, 1454 ^c^	1456 ^b,c,1^	1668 ^b^, 1667 ^c^	1664 ^b,c,2^	ND	0.7	0.5	ND	e, f
α-Humulene	1470 ^a^, 1467 ^c^, 1468 ^d^	1468 ^a,c,d,3^	1677 ^a^, 1682 ^c^, 1681 ^d^	1667 ^a,c,d,2^	10.1	ND	2.8	1.0	e, f, g
γ-Muurolene	1475 ^a^	1478 ^a,1^	1692 ^a^	1690 ^a,2^	1.2	ND	ND	ND	e, f
*trans*-9-*epi*-Caryophyllene	1477 ^b,d^	1466 ^b,1^	-	-	ND	0.4	ND	ND	e, f
Amorpha-4,7(11)-diene	1485 ^a^	1479 ^a,1^	1717 ^a^	-	1.6	ND	ND	ND	e, f
Germacrene D	1494 ^a,b^, 1493 ^c^	1481 ^a,b,c,2^	1713 ^a^, 1710 ^b^, 1721 ^c^	1708 ^a,b,c,2^	2.2	8.1	4.3	ND	e, f, g
β-Patchoulene	1395 ^d^	1388 ^d,5^	1497 ^d^	1488 ^d,3^	ND	ND	ND	1.7	e, f
β-Selinene	1503 ^a^	1490 ^a,1^	1722 ^a^	1717 ^a,2^	1.2	ND	ND	ND	e, f, g
α-Muurolene	1507 ^a^	1500 ^a,1^	1724 ^a^	1726 ^a,3^	1.3	ND	ND	ND	e, f
α-Selinene	1508 ^a^, 1507 ^d^	1498 ^a,d,1^	1726 ^a^, 1732 ^d^	1725 ^a,d,2^	0.9	ND	ND	0.2	e, f
Bicyclogermacrene	1511 ^b,d^	1500 ^b,1^	-	1734 ^b,2^	ND	0.4	ND	ND	e, f
δ-Cadinene	1526 ^a^	1523 ^a,2^	1755 ^a^	1756 ^a,2^	0.9	ND	ND	ND	e, f
cis-Calamenene	1529 ^a^	1529 ^a,1^	1828 ^a^	1835 ^a,2^	2.0	ND	ND	ND	e, f
N.I. M+• *m/z* 204 (Appendix A)	1532 ^a^	-	-	-	0.3	ND	ND	ND	-
α-Cadinene	1534 ^b^	1538 ^b,1^	-	1769 ^b,2^	ND	0.1	ND	ND	e, f
Caryophyllene oxide	1598 ^a^, 1595 ^c^, 1596 ^d^	1583 ^a,c,d,1^	1982 ^a^, 1993 ^c^, 1992 ^d^	1986 ^a,c,d,2^	3.8	ND	1.8	0.7	e, f, g
Guaiol	1607 ^a^	1600 ^a,1^	2083 ^a^	2089 ^a,2^	0.5	ND	ND	ND	e, f, g
Humulene epoxide II	1625 ^a^, 1624 ^d^	1608 ^a,d,1^	2036 ^a^, 2048 ^d^	2047 ^a,d,2^	1.3	ND	ND	0.1	e, f
γ-Eudesmol	1645 ^a^	1632 ^a,1^	2162 ^a^	2176 ^a,2^	0.6	ND	ND	ND	e, f
Coelution N.I. M+• *m/z* 220 + N.I. M+• *m/z* 204	1649 ^a^	-	-	-	0.6	ND	ND	ND	-
Caryophylla-4(12),8(13)-dien-5β-ol	1653 ^a^	1644 ^a,3^	-	-	0.8	ND	ND	ND	e, f
α-Cadinol	1668 ^a^	1654 ^a,1^	2224 ^a^	2227 ^a,2^	0.5	ND	ND	ND	e, f
α-Eudesmol	1671 ^a^	1653 ^1^	2213	2223 ^2^	1.5	ND	ND	ND	e, f
N.I. M+• *m/z* 222 (Appendix A)	1685 ^a^	-	2291 ^a^	-	4.5	ND	ND	ND	-
6-Methyl-hept-5-en-2-ona	985 ^c^	985 ^c,1^	1340 ^c^	1337 ^c,2^	ND	ND	1.3	ND	e, f, g
Citronellal	1152 ^c^	1153 ^c,1^	1482 ^c^	1475 ^c,2^	ND	ND	0.7	ND	e, f, g
Isocitral	1164 ^c^	1164 ^c,1^	-	-	ND	ND	1.6	ND	e, f
Isogeranial	1180 ^c^	1185 ^c,1^	1575 ^c^	-	ND	ND	2.1	ND	e, f
Nerol	1229 ^c^	1229 ^c,1^	1803 ^c^	1795 ^c,2^	ND	ND	2.5	ND	e, f, g
Neral	1246 ^c^	1242 ^c,2^	1692 ^c^	1679 ^c,2^	ND	ND	11.9	ND	e, f
Geraniol	1260 ^c^	1255 ^c,2^	1853 ^c^	1839 ^c,2^	ND	ND	19.0	ND	e, f, g
Neryl acetate	1376 ^c^	1361 ^c,1^	-	1718 ^c,2^	ND	ND	0.4	ND	e, f, g
Geranyl acetate	1383 ^c^	1381 ^c,1^	1756 ^c^	1751 ^c,2^	ND	ND	2.8	ND	e, f, g
α-Guaiene	1443 ^c^, 1449 ^d^	1437 ^c,1^, 1442 ^d,5^	1658 ^c^, 1607 ^d^	1652 ^c,2^, 1598 ^d,3^	ND	ND	2.2	13.3	e, f
Geranyl isobutanoate	1506 ^c^	1515 ^c,1^	1795 ^c^	1790 ^c,2^	ND	ND	1.2	ND	e, f
α-Bulnesene	1513 ^c^, 1514 ^d^	1509 ^c,d,1^	1730 ^c,d^	1729 ^c,d,4^	ND	ND	1.6	15.7	e, f
*trans*-α-Bisabolene	1545 ^c^	1544 ^c,3^	1777 ^c^	-	ND	ND	2.0	ND	e, f
Cycloseychellene	1430 ^d^	1418 ^d,3^	1579 ^d^	-	ND	ND	ND	0.7	e, f
α-Patchoulene	1479 ^d^	1461 ^d,5^	1656 ^d^	1649 ^d,3^	ND	ND	ND	6.3	e, f
Seychellene	1468 ^d^	1448 ^d,5^	1660 ^d^	1669 ^d,3^	ND	ND	ND	8.5	e, f
γ-Patchoulene	1485 ^d^	1441 ^d,3^	1674 ^d^	1656 ^d,3^	ND	ND	ND	1.3	e, f
N.I. M+• *m/z* 204 (Appendix A)	-	-	1689 ^d^	-	ND	ND	ND	0.5	-
δ-Selinene	1495 ^d^	1492 ^d,1^	1703 ^d^	-	ND	ND	ND	0.2	e, f
Aciphyllene	1505 ^d^	1501 ^d,1^	1713 ^d^	-	ND	ND	ND	2.9	e, f
γ-Gurjunene	1489 ^d^	1475 ^d,1^	1718 ^d^	1714 ^d,3^	ND	ND	ND	0.5	e, f
7-*epi*-α-Selinene	1533 ^d^	1534 ^d,3^	1773 ^d^	1764 ^d,2^	ND	ND	ND	0.2	e, f
Nootkatene	1519 ^d^	1517 ^d,1^	1786 ^d^	-	ND	ND	ND	0.1	e, f
N.I. M+• *m/z* 220 (Appendix A)	1579 ^d^	-	1898 ^d^	-	ND	ND	ND	0.2	-
N.I. M+• *m/z* 220 (Appendix A)	1583 ^d^	-	1944 ^d^	-	ND	ND	ND	0.6	-
N.I. M+• *m/z* 286 (Appendix A)	2112 ^c^	-	-	-	ND	ND	0.4	ND	e, f
Norpatchoulenol	1589 ^d^	1559 ^d,5^	2120 ^d^	-	ND	ND	ND	1.1	e, f
N.I. M+• *m/z* 222 (Appendix A)	-	-	2157 ^d^	-	ND	ND	ND	0.3	-
N.I. M+• *m/z* 222 (Appendix A)	1642 ^d^	-	2161 ^d^	-	ND	ND	ND	0.5	-
Patchoulol	1698 ^d^	1660 ^d,3^	2198 ^d^	2171 ^d,3^	ND	ND	ND	35.3	e, f
N.I. M+• *m/z* 206 (Appendix A)	-	-	2204 ^d^	-	ND	ND	ND	0.2	-
Pogostol	1679 ^d^	1655 ^d,3^	2217 ^d^	-	ND	ND	ND	3.2	e, f
Coelution N.I. M+• *m/z* 222 + N.I. M+• *m/z* 220	-	-	2261 ^d^	-	ND	ND	ND	0.2	-
Rotundone	1715 ^d^	1722 ^d,3^	2279 ^d^	-	ND	ND	ND	0.2	e, f
N.I. M+• *m/z* 220 (Appendix A)	-	-	2304 ^d^	-	ND	ND	ND	0.1	-
N.I. M+• *m/z* 222 (Appendix A)	1754 ^d^	-	2405 ^d^	-	ND	ND	ND	0.2	-
N.I. M+• *m/z* 218 (Appendix A)	1727 ^d^	-	2414 ^d^	-	ND	ND	ND	0.1	-
Dehydrofukinone	1823 ^d^	1820 ^d,3^	2467 ^d^	2404 ^d,3^	ND	ND	ND	0.1	e, f
N.I. M+• *m/z* 218 (Appendix A)	1802 ^d^	-	2496 ^d^	-	ND	ND	ND	0.1	-
Pogostone	1724 ^d^	-	2576 ^d^	-	ND	ND	ND	1.1	e

Linear retention indices (LRI): ^a^. *L. origanoides*, phellandrene chemotype EO; ^b^. *L. alba*, carvone chemotype EO; ^c^. *L. alba* citral chemotype EO; ^d^. *P. cablin* EO; e. Tentative identification based on mass spectra (MS, EI, 70 eV, coincidence > 90%); f. Identification of the compound based on LRI, measured in columns DB-WAX and DB-5MS; g. Confirmatory identification by MS and LRI, using reference (standard) compound. ^1^ Adams [14]; ^2^ Babushok et al. [18]; ^3^ NIST [15]; ^4^ Davies [19]; ^5^ Van Beek and Joulain [20]. ND. Non-detected; -. Non-available information.

**Table 2 insects-14-00041-t002:** Percentage repellency (PR) and mean repellent concentration value (RC_50_) for EOs and DEET against *U. dermestoides* after two exposure times.

EO/Commercial Repellent	Concentration (μL/mL)	% Repellency Obtained after Different Exposure Times (Hours)	
		2	4
*L. origanoides* (phellandrene-rich chemotype)	2	18 ± 8	15 ± 2
	4	20 ± 7 a	20 ± 8 a
	8	73 ± 6 a	68 ± 10 a
	16	100 ± 0 a	100 ± 0 a
	RC_50_ (%)	4.2 (3.9–4.4) *	4.3(4.1–4.5)
	Chi-square (X^2^), *p*-value	49.70, *p* < 0.0001	49.22, *p* < 0.0001
*L. alba* (carvone-rich chemotype)	2	83 ± 6 a	60 ± 8 a
	4	85 ± 8 a	80 ± 10 a
	8	90 ± 5 a	88 ± 8 a
	16	100 ± 0 a	98 ± 3 a
	RC_50_ (%)	1.5 (1.4–1.7)	2.2 (2.1–2.4)
	Chi-square (X^2^), *p*-value	6.573, *p* = 0.0104	16.95, *p* < 0.0001
*L. alba* (citral-rich chemotype)	2	85 ± 5 a	75 ± 4 a
	4	98 ± 3 a	93 ± 4 a
	8	100 ± 0 a	98 ± 3 a
	16	100 ± 0 a	100 ± 0 a
	RC_50_ (%)	1.0 (0.9–1.1)	1.5 (1.4–1.6)
	Chi-square (X^2^), *p*-value	10.54, *p* = 0.0012	15.30, *p* < 0.0001
*P. cablin*	2	75 ± 7 a	70 ± 10 a
	4	95 ± 3 a	95 ± 3 a
	8	98 ± 3 a	98 ± 3 a
	16	100 ± 0 a	100 ± 0 a
	RC_50_ (%)	1.4 (1.3–1.6)	1.6 (1.4–1.7)
	Chi-square (X^2^), *p*-value	15.41, *p* < 0.0001	19.15, *p* < 0.0001
DEET	2	–13 ± 11	–25 ± 13 b
	4	38 ± 8 a	23 ± 13
	8	75 ± 7 a	60 ± 5 a
	16	90 ± 5 a	73 ± 8 ab
	RC_50_ (%)	5.9 (5.7–6.1)	9.1 (8.6–9.6)
	Chi-square (X^2^), *p*-value	61.48, *p* < 0.0001	48.70, *p* < 0.0001

a. Statistically significant differences between the number of organisms in the treated and untreated area, paired *t*-test (*p* < 0.05). b. Statistically significant differences between PRs when the same tested concentrations were compared at 2 and 4 h exposure, paired *t*-test (*p* < 0.05). *. 95% confidence intervals.

**Table 3 insects-14-00041-t003:** Preference index (PI) for insects treated with essential oils or commercial repellent in different concentrations after two exposure times.

EO/Commercial Repellent	Concentration (μL/mL)	Preference Index *	
		2 h	4 h
*L. origanoides* (phellandrene chemotype)	2	−0.18 ± 0.08 R	−0.15 ± 0.15 R
	4	−0.20 ± 0.07 R	−0.20 ± 0.08 R
	8	−0.73 ± 0.06 R	−0.68 ± 0.10 R
	16	−1.00 ± 0.00 R	−1.00 ± 0.00 R
*L. alba* (carvone chemotype)	2	−0.83 ± 0.06 R	−0.60 ± 0.08 R
	4	−0.85 ± 0.08 R	−0.80 ± 0.11 R
	8	−0.90 ± 0.05 R	−0.88 ± 0.08 R
	16	−1.00 ± 0.00 R	−0.98 ± 0.03 R
*L. alba* (citral chemotype)	2	−0.85 ± 0.05 R	−0.75 ± 0.03 R
	4	−0.98 ± 0.03 R	−0.93 ± 0.04 R
	8	−1.00 ± 0.00 R	−0.98 ± 0.03 R
	16	−1.00 ± 0.00 R	−1.00 ± 0.00 R
*P. cablin*	2	−0.75 ± 0.07 R	−0.70 ± 0.10 R
	4	−0.95 ± 0.03 R	−0.95 ± 0.03 R
	8	−0.98 ± 0.03 R	−0.98 ± 0.03 R
	16	−1.00 ± 0.00 R	−1.00 ± 0.00 R
DEET	2	0.13 ± 0.11 A	0.25 ± 0.13 A
	4	−0.38 ± 0.08 R	−0.23 ± 0.13 R
	8	−0.75 ± 0.07 R	−0.60 ± 0.05 R
	16	−0.90 ± 0.05 R	−0.73 ± 0.07 R

*. Preference index (PI); Rating: R = repellent and A = attractive.

## Data Availability

All datasets used in this study can be provided by the authors upon reasonable request.

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
