# Peer review of "Chemical Composition, Repellent Action, and Toxicity of Essential Oils from Lippia origanoide, Lippia. alba Chemotypes, and Pogostemon cablin on Adults of Ulomoides dermestoides (Coleoptera: Tenebrionidae)"

_insects, 2022, doi:10.3390/insects14010041_

Round 1

Reviewer 1 Report

The manuscript is about the chemical composition, repellency and contact toxicity of four different essential oils against Ulomoides dermestoides. It is well written and just need some revision before deciding about acceptance.

My comments are as below:

Line 19: please change “several” to “some” or “four” essential oils…

Line 21:  trans-β-cariofilene is correct

Line 71: use “U. dermestoides” instead of “Ulomoides dermestoides

Line 128, 129 and in all of manuscript: use “treatment” instead of “vehicle”.

Line 131: use “filter paper” instead of “filter”

Line 131-132: did you use adult insect in the same age for your test? Or selected them by chance from different ages in the colony? please mention this in the materials and methods.

Line 133: was there a laboratory condition for repellency test? Or it was done in room temperature? Please mention it.

Line 143: treatment is ok. Please delete “ vehicle”

Line 145: How many unsexed adults?

Line 147: change “ animals” to “insects”.

In table 2: How did you calculate RC50? I couldn’t find explanation about that in materials and methods. Please add explanations about RC50

LINE 230: please add “ and” before “ fumigant”

Line 260: use “ contact toxicity” instead of “toxicity”.

The author can use the below reference which is similar to the manuscript for comparing the result or/and use it for methodology and discussion.

Zandi-Sohani, N., Hojjati, M., & Carbonell-Barrachina, Á. A. (2012). Bioactivity of Lantana camara L. essential oil against Callosobruchus maculatus (Fabricius). Chilean journal of agricultural research72(4), 502.

Author Response

Reviewer 1

The manuscript is about the chemical composition, repellency and contact toxicity of four different essential oils against Ulomoides dermestoides. It is well written and just need some revision before deciding about acceptance.

My comments are as below:

Line 19: please change “several” to “some” or “four” essential oils…

Answer: Thanks for your suggestion. The word “several” was changed to “four”

Line 21:  trans-β-cariofilene is correct.

Answer: Thanks for your suggestion. The name of the compound has been corrected.

Line 71: use “U. dermestoides” instead of “Ulomoides dermestoides

Answer: The scientific name “Ulomoides dermestoides” was changed to “U. dermestoides

Line 128, 129 and in all of manuscript: use “treatment” instead of “vehicle”.

Answer: The word “vehicle” has been deleted in the manuscript.

Line 131: use “filter paper” instead of “filter”

Answer: The word “filter” was changed to “filter paper”

Line 131-132: did you use adult insect in the same age for your test? Or selected them by chance from different ages in the colony? please mention this in the materials and methods.

Answer:  Thanks for your observation. The sentence “4-8 days old, healthy adults were randomly chosen for bioassays” has been included.

Line 133: was there a laboratory condition for repellency test? Or it was done in room temperature? Please mention it.

Answer:  Thanks for your observation. The sentence “under standard laboratory conditions of 28 °C (±1 °C), a 10:14 h light:dark photoperiod and relative humidity of 70-85%” has been included.

Line 143: treatment is ok. Please delete “vehicle”

Answer:  Thanks for your observation. The word “vehicle” has been deleted in the manuscript.

Line 145: How many unsexed adults?

Answer:  Thanks for your observation. The word “ten” has been  included

Line 147: change “animals” to “insects”.

Answer: Thanks for your suggestion. The word “animals” was changed to “insects”

In table 2: How did you calculate RC50? I couldn’t find explanation about that in materials and methods. Please add explanations about RC50

Answer:  Thanks for your observation. The following paragraph was added in the data analysis section:

“Data obtained from each concentration-repellence bioassay were subjected to Probit analysis, and 50% repellence concentrations (RC50) was determined by log-Probit regression”

LINE 230: please add “and” before “fumigant”

Answer: Thanks for your suggestion. The word “and” was added.

Line 260: use “contact toxicity” instead of “toxicity”.

Answer: Thanks for your suggestion. The word “toxicity” was changed to “contact toxicity”

The author can use the below reference which is similar to the manuscript for comparing the result or/and use it for methodology and discussion.

Zandi-Sohani, N., Hojjati, M., & Carbonell-Barrachina, Á. A. (2012). Bioactivity of Lantana camara L. essential oil against Callosobruchus maculatus (Fabricius). Chilean journal of agricultural research72(4), 502.

Answer: Thanks for your suggestion. The reference “Zandi-Sohani et al., 2012” was added.

Reviewer 2 Report

The manuscript entitled “Chemical composition, repellent action and toxicity of essential oils from Lippia origanoide, Lippia. alba chemotypes and Pogostemon cablin on adults of Ulomoides dermestoides (Coleoptera: Tenebrionidae)” brings new and interesting information on the repellent and toxicity activity of the extract of Lippia origanoide, Lippia. alba chemotypes and Pogostemon cablin. Manuscript looks well formatted according to journal guidelines. It is well written in general. Conclusions are supported by the results but I suggest some minor changes in Figure formats. I recommend this paper for publication after considering the minor points showed in the attached marked pdf.

Author Response

The manuscript entitled “Chemical composition, repellent action and toxicity of essential oils from Lippia origanoideLippia. alba chemotypes and Pogostemon cablin on adults of Ulomoides dermestoides (Coleoptera: Tenebrionidae)” brings new and interesting information on the repellent and toxicity activity of the extract of Lippia origanoideLippia. alba chemotypes and Pogostemon cablin. Manuscript looks well formatted according to journal guidelines. It is well written in general. Conclusions are supported by the results but I suggest some minor changes in Figure formats.

I recommend this paper for publication after considering the minor points showed in the attached marked pdf.

Answer: Thanks for your suggestion. Corrections have been made in the manuscript.

This conclusion is not appropriate for this study, needs to be rewritten. “These findings show that the EOs may prove novel active principles for formulations of repellents to prevent and control insect infestations of stored products”

Answer:

Thanks a lot for your comment.

We added the following sentence:  These findings show that the EOs provide active molecules for natural repellent formulations to prevent and control insect infestations of stored products. 

Introduction need an implementation with a better focus on the target species and some more background on the phellandrene-rich L. origanoides, carvone-rich L. alba, citral-rich L. alba, and P. cablin. Give some detail about these plants and why these plants should have some behavioral effect on the peanut beetle.

Answer: Thanks for your suggestion. The following paragraph has been added “In the others hand, there is an interest in studying and evaluating the use of EOs for pest management as a result of insect resistant to the synthetic insecticides. In fact, various studies have investigated the bioactivity of EOs and their potential uses as bio-insecticides against insect pests. In this sense, EOs from Lippia origanoides (phel-landrene), L. alba (carvone) and Pogostemon cablin have shown repellent and insecticidal properties against Aedes aegypti [10], Sitophilus zeamais [11], Tribolium castaneum [11, 12].”

Line 81. This part should be more stressed, giving more details “2.2. Plant material and extraction of EOs

Answer: Thanks for your suggestion. Corrections have been made in the manuscript, as follows: “EOs of Lippia origanoides, phellandrene-rich chemotype, Lippia alba, carvone-rich chemotype, Lippia alba, citral-rich chemotype and Pogostemon cablin were obtained from aerial parts using steam distillation in a 0.4 m3 stainless steel column and separated by decantation. Subsequently, they were dried with Na2SO4 and stored at 4 â—¦C in amber flasks”.

Line 140. Four replicates are not enough for statistics here, better less individuals and more replications.

Answer:  

The reviewer has a good point.  However, all groups had 10 individuals, this means that in total we had 8 different counts from two different experiments.  We will follow the reviewer’s advice on future experiments.

Line 149. These replicates are not enough for statistics here, better less individuals and more replications.

Answer:  

The reviewer has a good point.  However, all groups had 10 individuals, this means that in total we had 8 different counts from two different experiments.  We will follow the reviewer’s advice on future experiments.

Figure 1. I suggest using the same Y scale (100%) for the the figures. Y scale should be corrected mortality. Also there are some ambiguity of X scale in Figure 1.

Thanks for your suggestion. The figure 1 has been changed according to your suggestion.

Round 2

Reviewer 2 Report

The authors have improved the manuscript well.